# The diagnostic value of core needle biopsy in cervical cancer: A retrospective analysis

**Massimiliano Lia** [1]*, **Lars-Christian Horn**[2], **Paulina Sodeikat**[1], **Michael Höckel**[1], **Bahriye Aktas** [1], **Benjamin Wolf**[1]

**1** Department of Gynecology and Obstetrics, University Hospital Leipzig, Leipzig, Germany, **2** Division of Gynecologic, Breast and Perinatal Pathology, University Hospital Leipzig, Leipzig, Germany

* massimiliano.lia@medizin.uni-leipzig.de

**Data Availability Statement:** A minimal anonymized data set, including data about the tumor and the CNB (tumor characteristics, stage, MRI-data, number of CNB) can be found within the Supporting Information. The study protocol and its

## Abstract

Cervical carcinoma is a major cause of morbidity and mortality among women worldwide. Histological subtype, lymphovascular space invasion and tumor grade could have a prognostic and predictive value for patients' outcome and the knowledge of these histologic characteristics may influence clinical decision making. However, studies evaluating the diagnostic value of various biopsy techniques regarding these parameters of cervical cancer are scarce. We reviewed 318 cases of cervical carcinoma with available pathology reports from preoperative core needle biopsy (CNB) assessment and from final postoperative evaluation of the hysterectomy specimen. Setting the postoperative comprehensive pathological evaluation as reference, we analysed CNB assessment of histological tumor characteristics. In addition, we performed multivariable logistic regression to identify factors influencing the accuracy in identifying LVSI and tumor grade. CNB was highly accurate in discriminating histological subtype. Sensitivity and specificity were 98.8% and 89% for squamous cell carcinoma, 92.9% and 96.6% for adenocarcinoma, 33.3% and 100% in adenosquamous carcinoma respectively. Neuroendocrine carcinoma was always recognized correctly. The accuracy of the prediction of LVSI was 61.9% and was positively influenced by tumor size in preoperative magnetic resonance imaging and negatively influenced by strong peritumoral inflammation. High tumor grade (G3) was diagnosed accurately in 73.9% of cases and was influenced by histological tumor type. In conclusion, CNB is an accurate sampling technique for histological classification of cervical cancer and represents a reasonable alternative to other biopsy techniques.

## Introduction

Cervical carcinoma is a major cause of disability, morbidity, and mortality among women with an estimated number of worldwide deaths of 311,000 in 2018 [1].

In patients diagnosed with cervical cancer the treatment strategy depends on locoregional and distant disease extent. Several pathological characteristics have been studied to predict advanced disease (e.g. lymph node metastasis), poor prognosis, or disease recurrence. Of

amendments were approved by the ethics committee of the University of Leipzig (Stephanstraße 9A.1, 04103 Leipzig, Germany). The sharing of raw data is not supported by the ethical committee and thus not included in the written consent form of the study patients.

**Funding:** The non-profit-organization "Stiftung gynäkologische Onkologie" granted payments to authors of this study (MH & BW) and provided funding for the data collection. This organization had no influence on study design, data analysis, manuscript preparation or the decision to publish this study.

**Competing interests:** I have read the journal's policy and the authors of this manuscript have the following competing interests: MH and BW received payments from the non-profit-organization "Stiftung gynäkologische Onkologie". BA received honoraria from Pfizer Inc, Roche AG, Novartis AG, AstraZeneca PLC, Amgen Inc and Daiichi Sankyo Co Ltd. LCH, PS and ML have no conflicts to disclose. This does not alter our adherence to PLOS ONE policies on sharing data and materials.

**Abbreviations:** AC, adenocarcinoma; aOR, adjusted odds ratio; CNB, core needle biopsy; LVSI, lymphovascular space invasion; MRI, magnetic resonance imaging; SCC, squamous cell carcinoma.

these, the most commonly reported prognostic factors are lymphovascular space invasion (LVSI) and tumor grade.

However, most studies examine conization, trachelectomy or radical hysterectomy specimens and few authors have focused on preoperative biopsy samples in predicting the tumor grade or LVSI in cervical cancer. Pre-treatment biopsy specimens are frequently the only tissue samples available for analysis, since many patients do not undergo surgery due to national and international guidelines which generally recommend primary chemo-radiotherapy for locally advanced cervical cancer, i.e. tumors staged IB2, IIA2, IIB, IIIB and IVA according to the Fédération Internationale de Gynécologie et d'Obstetrique (FIGO) 2009 criteria [2, 3]. Knowledge about how histological characteristics assessed in different biopsy techniques correlate with those in final hysterectomy specimens is therefore important if clinical decision making is based on these factors.

Core needle biopsy (CNB) is a safe and accurate way to obtain tissue specimens and has been studied in the preoperative evaluation of abnormalities of the breast [4], peripheral nerve sheath tumors [5], thyroid [6] and pulmonary nodules [7]. In pelvic lesions suspicious of gynecologic malignancy it has been shown that this biopsy technique (with ultrasound-guidance) is safe, yields adequate tissue samples and almost always provides a reliable diagnosis [8, 9]. However, it is unknown if CNB can accurately predict specific tumor characteristics (cancer type, LVSI, tumor grade) in gynecologic malignancies.

This study aims to evaluate the performance of preoperative CNB in cervical carcinoma.

## Material and methods

This investigation represents a retrospective subgroup analysis of patients enrolled in the prospective observational Leipzig School Mesometrial Resection (MMR) study. All patients were cared for at the Department of Gynecology at the University Hospital Leipzig and were treated by total or extended mesometrial resection (TMMR, EMMR), or by laterally extended endopelvic resection (LEER). These are surgical treatments for cervical cancer based on the theory of ontogenetic cancer fields. The study outcomes along with a detailed description of the techniques have been published [10–14].

All consecutive patients who presented to our institution with primary cervical cancer staged FIGO (Féderation Internationale de Gynécologie et d'Obstetrique) IB1 –IIB and who were older than 18 years of age were eligible for inclusion. In addition, selected patients with cancer staged FIGO IIIA, IIIB, and IVA were included in the trial if they were not candidates for primary (chemo-)radiotherapy or declined such treatment and insisted on surgery. Patients were excluded if they were not fit for surgical treatment. The study protocol and its amendments were approved by the ethics committee of the University of Leipzig (012/13-28012013, 171–2006, 192/2001, and 151/2000) and are registered with the German Clinical Trials Register (DRKS00015171). At the time of enrollment in the MMR study, all patients provided informed written consent which included further usage of study data.

For this current analysis, a computer search of the study database was performed and the medical records of MMR-study-patients with primary cervical cancer between December of 1999 and May of 2017 were retrospectively analysed. Patients were excluded if they had received preoperative chemotherapy, if pathological reports were not available and if the CNB did not contain any tumor tissue. If only single histopathological characteristics (e.g. tumor grade, peritumoral inflammation) were not available, the patient was excluded from analyses involving this missing characteristic (but not for others).

As specified in the MMR study protocol, all patients had undergone preoperative staging diagnostics (using the FIGO-classification of 2009) which always included an examination

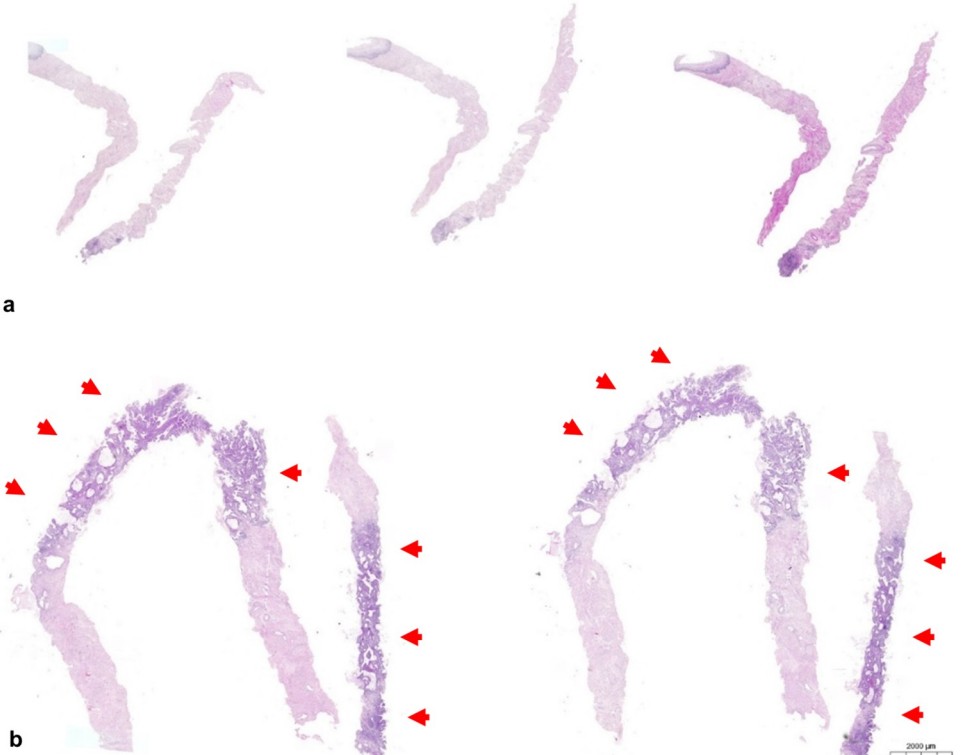

**Fig 1. Histologic examination of core needle biopsies.** A case representing two core needle biopsies examined using three step sections in (a). Core needle biopsy in (b) representing three cores showing tumor infiltration from 20% to 80% (arrows).

under general anesthesia during which core needle biopsies of the cervix were taken using an automated spring-loaded biopsy device (Bard Magnum® Biopty Gun, Bard, UK) with a 14 Gauge needle. The number of biopsies performed was decided by the examiner in order to ensure a proper tissue sample for a complete histologic classification of the tumor.

The core needle biopsies were completely processed with embedding of multiple cores within one cassette in cases where multiple cores were obtained. From each block three step sections were performed with intervals of about 200μm between the steps (Fig 1). No immuno-histochemical stains were routinely performed for establishing the diagnosis. Both the core needle biopsies and the hysterectomy specimens were analysed under the supervision of a dedicated gynecologic pathologist (LCH) and the reports were reviewed for this study. Specimens were examined for histologic tumor type, (conventional) tumor grade, lymphovascular space invasion, and peritumoral inflammatory response (Fig 2). Peritumoral inflammatory response was classified according to previous publications as absent, weak, moderate or strong [15]. Additionally, conventional tumor grading was performed as described before [16, 17] and G1- and G2-tumors were merged into one group thus adopting the binary grading model suggested by Horn et al. [18]. These data were included in the database and compared with the same pathological parameters of the subsequent surgical specimens.

All patients underwent magnetic resonance imaging (MRI) as part of the preoperative staging procedures. Both tumor size and suspected parametrial involvement were included into the analysis.

In order to assess the performance of CNB, we calculated the sensitivity, specificity, accuracy, positive (PPV) and negative (NPV) predictive value for every pathological parameter and

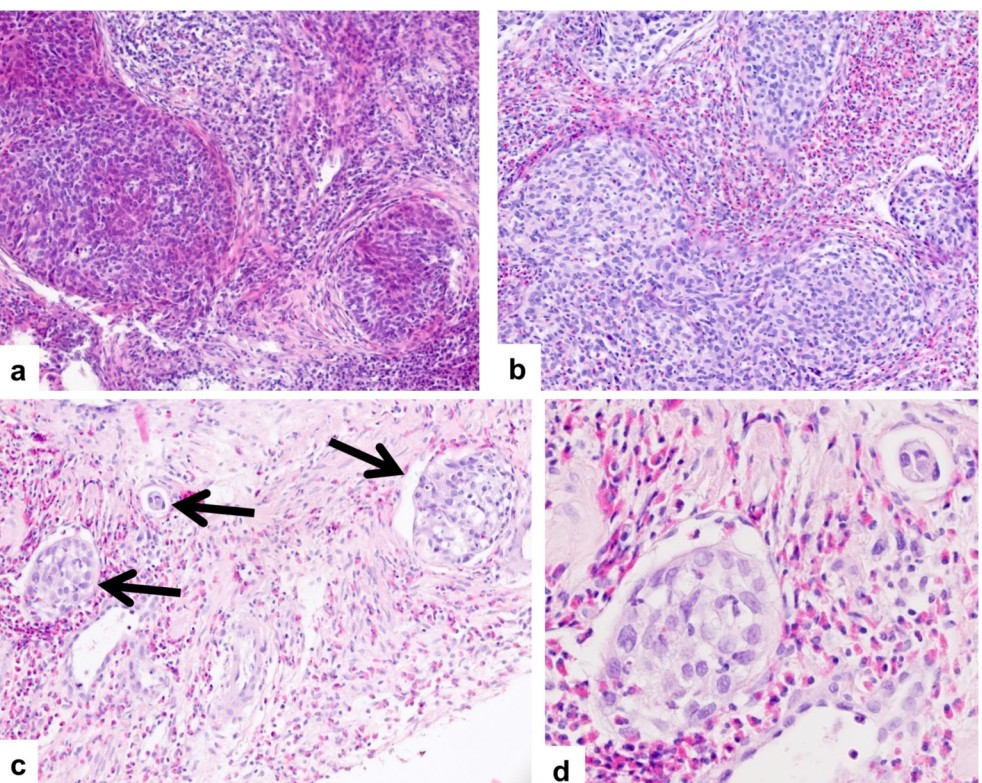

**Fig 2. Peritumoral inflammation and lymphovascular space invasion.** Squamous cell carcinoma with strong peritumoral inflammtory response with lymphocytic (a) and eosinophilic (b) predominance. Multiple lymphovascular involvement (arrows) is shown in (c). Intravascular tumor cell surrounded by lymphoepithelial cells in lymphovascular space invasion can be seen in (d).

used Fisher's exact test to compare sensitivities and specificities. To quantify the concordance between the pathology findings in CNB and hysterectomy specimens we used the (unweighted) Cohen´s kappa. To establish agreement quality for each kappa-value we applied the classification suggested by Landis and Koch. In this classification kappa-values of less than 0.2 correspond to a slight agreement, 0.21–0.4 to a fair agreement, 0.41–0.6 to a moderate agreement, 0.61–0.8 to a substantial agreement and 0.81–1.0 to an almost perfect agreement [19].

Furthermore, we performed a multivariable logistic regression analysis to investigate whether additional variables influenced CNB accuracy and to compute the adjusted odds ratios (aOR) of these variables. As previously proposed, variables were included in the multi-variable logistic regression if they were associated with these parameters in an univariable model with a p-value of <0.25 [20]. We ensured that events per variable were always more than 10 in multivariable logistic regression analysis in order to avoid bias of the coefficients [21]. P-values of <0.05 were regarded as statistically significant.

Statistical analysis was performed with the program R [22]. Kappa statistics were computed with the "IRR"-package. The most appropriate thresholds of continuous variables (specifically tumor size in the MRI) for the logistic regression were computed with ROC-curve analysis using the "pROC"-package. The "blorr"-package was used to perform collinearity diagnostics. Categorical data is given as percentages while continuous data is reported as numbers.

## Results

The reports of 458 patients were identified between December of 1999 and May of 2017. In six cases CNB had not been performed (e.g. biopsies had already been performed before the examination under general anesthesia). Further 134 cases were excluded, since no tumor was detected in the CNB specimen. This false negative results of the CNB could partially be explained by the fact that 73.1% of these patients had a conization prior to the examination under general anesthesia, thus removing a significant part of the tumor mass before CNBs could be performed. Eventually, the reports of 318 patients were included in the study.

Histologic tumor type, LVSI and tumor grade were known in all 318 patients in both CNB and hysterectomy specimen. Peritumoral inflammation could be evaluated in 284 CNBs and 290 hysterectomy specimen. Patient and tumor characteristics are compiled in Table 1.

The sensitivity of CNB in recognizing a squamous cell carcinoma (SCC) and adenocarcinoma (AC) as such was 98.8% and 92.9%, respectively. Neuroendocrine carcinoma was recognized correctly by CNB in both cases. There were 15 cases of adenosquamous cervical carcinoma in our study group. The CNB recognized five of them correctly (33.3%), but classified four (26.7%) as SCC and six (40%) as AC. Specificity of the CNB for the various histological types was 89% for SCC, 96.6% for AC, and 100% for adenosquamous and neuroendocrine cervical carcinoma. Overall, the CNB accurately predicted histologic tumor type in 94.7% of cases.

The overall sensitivity and specificity for the detection of LVSI in CNB were 56% and 83.8% respectively. The accuracy corresponded to 61.9% (kappa 0.26). PPV and NPV were 92.7% and 34.1% respectively.

Univariable logistic regression showed that presence of strong peritumoral inflammation, advanced clinical stage (FIGO IIB and higher), parametrial involvement on MRI, tumor size over 3.8 cm on MRI and number of CNB were associated with correct LVSI assessment. In multivariable logistic regression the presence of strong peritumoral inflammation and a tumor size larger than 3.8 cm had a statistically significant influence on correct LVSI assessment (Table 2).

A tumor size of more than 3.8 cm as determined by MRI was associated with an increase of an accurate diagnosis (aOR 2.1, 95% confidence interval [CI]: 1.11–4.03).

The presence of strong peritumoral inflammation negatively influenced correct LVSI status prediction of the CNB (aOR 0.46, 95% CI: 0.22–0.95)). This decrease in accuracy was predominantly caused by a significantly lower sensitivity, which decreased from 63.2% to 38.5% when strong peritumoral inflammation was present (p = 0.0067). Conversely, specificity didn't change significantly (80.6% vs. 84.6%, p = 1.0).

When analysing the performance of CNB predicting tumor grade we adopted the dual classification suggested by Horn et al. [18] subdividing tumor grades in high grade (G3) and low grade (G1-2). The overall sensitivity and specificity of CNB predicting a high-grade (G3) tumor were 53% and 89.7% respectively. The accuracy corresponded to 73.9% (kappa 0.45). PPV and NPV were 78.9% and 72.4% respectively.

Univariable logistic regression showed that strong peritumoral inflammation, LVSI, advanced clinical stage (FIGO IIB and higher), parametrial involvement on MRI, and post-conization status were associated with correct diagnosis of a high-grade-tumor through CNB. Also we observed that, in the univariable model, SCC in the CNB influenced tumor grade recognition negatively (OR 0.43, 95% CI: 0.2–0.85, p = 0.021) while AC improved it (OR 2.29, 95% CI: 1.12–5.19, p = 0.032). Since these two variables were highly collinear, we only included the variable describing the presence of SCC in our multivariable model, since this seemed to have the highest influence on tumor grade recognition. Multivariable logistic regression

**Table 1. Patient- and tumor characteristics.**

| | | | |
|---|---|---|---|
| | | Age—years (median, IQR) | 45.5 (37–55.75) |
| | | Preoperative conization—no. (%) | 56 (17.6%) |
| | | Suspected parametrial involvement (MRI)—no. (%) | 121 (40.4%) |
| | | Tumor size (MRI)—cm (median, IQR) | 3.7 (2.5–4.6) |
| Histologic subtype | | Squamous cell carcinoma | 245 (77.1%) |
| n = 318 | | Adenocarcinoma | 56 (17.6%) |
| | | Adenosquamous carcinoma | 15 (4.7%) |
| | | Neuroendocrine carcinoma | 2 (0.6%) |
| Stage of disease (FIGO) no. (%) | | IA | 3 (0.9%) |
| n = 318 | | IB1 | 117 (36.8%) |
| | | IB2 | 34 (10.7%) |
| | | IIA | 27 (8.5%) |
| | | IIB | 118 (37.1%) |
| | | IIIA | 2 (0.6%) |
| | | IIIB | 11 (3.5%) |
| | | IV | 6 (1.9%) |
| Histological tumor stage | | pT1b1 | 109 (34.3%) |
| n = 318 | | pT1b2 | 46 (14.5%) |
| | | pT2a1 | 10 (3.1%) |
| | | pT2a2 | 5 (1.6%) |
| | | pT2b | 140 (44.0%) |
| | | pT3b | 2 (0.6%) |
| | | pT4 | 6 (1.9%) |
| Lymphovascular space invasion (LVSI) present | CNB | | 151 (47.5%) |
| n = 318 | Hysterectomy | | 250 (78.6%) |
| Tumor grading | CNB | G1-2 (low-grade carcinoma) | 228 (71.7%) |
| n = 318 | | G3 (high-grade carcinoma) | 90 (28.3%) |
| | Hysterectomy | G1-2 (low-grade carcinoma) | 183 (57.5%) |
| | | G3 (high-grade carcinoma) | 135 (42.5%) |
| Peritumoral inflammation | CNB | absent | 44 (15.5%) |
| n = 284 | | mild | 113 (39.8%) |
| | | moderate | 75 (26.4%) |
| | | severe | 52 (18.3%) |
| n = 290 | Hysterectomy | absent | 57 (19.7%) |
| | | mild | 95 (32.7%) |
| | | moderate | 75 (25.9%) |
| | | severe | 63 (21.7%) |
| Number of CNBs | 1 | | 26 (8.2%) |
| n = 318 | 2 | | 74 (24.8%) |
| mean = 3.4 | 3 | | 98 (30.8%) |
| | 4 | | 54 (17.0%) |
| | 5 | | 17 (5.3%) |
| | 6 | | 24 (7.6%) |
| | > = 7 | | 20 (6.3%) |

IQR = interquantile range.

MRI = magnetic resonance imaging.

**Table 2. Multivariable logistic regression model for correct assessment of lymphovascular space invasion (LVSI) and tumor grading.**

| Multivariable logistic regression model for correct assessment of lymphovascular space invasion (LVSI) | | | | |
|---|---|---|---|---|
| | Estimate | Std. Error | P-Value | Adjusted Odds Ratio (95% CI) |
| Presence of strong peritumoral inflammation in CNB | -0.78203 | 0.37077 | 0.0349 | 0.46 (0.22–0.95) |
| Tumor size on MRI (> 38mm) | 0.7388 | 0.32902 | 0.0247 | 2.1 (1.11–4.03) |
| Suspected parametrial involvement on MRI | 0.0548 | 0.34451 | 0.8736 | 1.06 (0.54–2.08) |
| Advanced disease (FIGO ≥ IIB) | 0.22483 | 0.34202 | 0.511 | 1.25 (0.64–2.46) |
| Number of CNBs | 0.06241 | 0.08248 | 0.4492 | 1.06 (0.91–1.26) |
| Multivariable logistic regression model for correct assessment of tumor grading | | | | |
| | Estimate | Std. Error | P-Value | Adjusted Odds Ratio (95% CI) |
| Squamous cellular cancer in CNB | -1.18347 | 0.50474 | 0.019 | 0.31 (0.1–0.76) |
| Conization performed prior staging | 0.71038 | 0.4519 | 0.116 | 2 (0.88–5.31) |
| LVSI in CNB | -0.4082 | 0.31023 | 0.188 | 0.66 (0.36–1.22) |
| Presence of strong peritumoral inflammation in CNB | -0.47959 | 0.37603 | 0.2 | 0.62 (0.3–1.31) |
| Advanced disease (FIGO ≥ IIB) | 0.01439 | 0.34175 | 0.966 | 1.01 (0.52–2.0) |
| Suspected parametrial involvement on MRI | -0.41518 | 0.33771 | 0.219 | 0.66 (0.34–1.28) |

MRI = magnetic resonance imaging.

CNB = core needle biopsy.

LVSI = lympho-vascular space invasion.

(Table 2) concluded that SCC-histology in the CNB impaired correct grading (aOR 0.31, 95% CI: 0.1–0.76).

## Discussion

In this study, we show that CNB has high accuracy in discriminating the major histologic subtypes of cervical carcinoma. The sensitivity and specificity of CNB in recognizing the most common histologic tumor types were high (SCC: 98.8% and 89% respectively; AC: 92.9% and 96.6% respectively). Not surprisingly, adenosquamous cervical carcinomas were misclassified in 66.7% of cases (10 out of 15) as either SCC or AC. Both cases of neuroendocrine carcinoma were identified as such by the CNB.

In the only study evaluating the performance of superficial cervical biopsies, Bidus et al. found that this technique had a sensitivity, specificity, NPV and PPV of 14%, 96%, 45% and 83% in the detection of LVSI respectively [23]. In our study, sensitivity was considerably higher (56%) while specificity was seemingly slightly lower (83.8%). Predictive values however are difficult to compare, since the study of Bidus and colleagues had a considerably lower prevalence of LVSI due to the fact that most cases were early cervical cancers.

While data regarding superficial biopsies of cervical cancer are scarce, the accuracy of excision specimen regarding the detection of LVSI has been studied by various authors. Bai et al. and Kim et al. studied conization in stage IA2 –IB1 cervical cancers [24, 25] while Bidus et al. also included stage IB2 –IIA (7.1% of cases) [23]. In these studies, sensitivity and specificity of conization ranged from 37.5–70.5% and 80–88% respectively. Our data show that CNB has a sensitivity of 56% and a specificity of 83.8%, suggesting similar performance in recognizing LVSI compared with conization. Interestingly, PPV and NPV of conization ranged between 43–75% and 57–90% respectively [23, 25] while those of CNB differed substantially in our study. We observed a PPV and NPV of 92.7% and 34.1% respectively for this sampling technique. However, it should be taken into account that prevalence of LVSI, which has a major influence on its predictive values, was very high in our study (78.6%) because of a high

proportion of advanced cervical carcinomas (Table 1). In contrast, those studies evaluating the accuracy of conization observed LVSI in 15.8–18.8% of their cases [23–25]. The kappa value for LVSI assessment in our study was 0.26 (fair agreement [19]). In summary, our data suggest that CNB may have a similar accuracy to conization in recognizing LVSI. Additionally, the drop in sensitivity for detection of LVSI caused by strong peritumoral inflammation has not been described in cervical cancer so far and warrants further investigation.

LVSI has been associated with parametrial [26–28] and vaginal [28] involvement, lymph node metastasis [29–31], and disease recurrence [32]. However, it has also been observed that LVSI is heterogeneous and that diffuse LVSI was associated with a worse prognosis while focal LVSI did not significantly influence disease-free survival [33]. This may explain why some studies failed to find a significant impact of LVSI on patients' prognosis [34]. Nevertheless, various authors have suggested that LVSI should play a role in treatment choice such as in the decision for adjuvant radiotherapy [35, 36], type of radical hysterectomy [37] or the need for lymph node assessment in early cervical cancer [38].

CNB showed an accuracy of 73.9% regarding the correct distinction between low-grade (G1 and G2) and high-grade (G3) tumors with a kappa of 0.45 (moderate agreement [19]). This accuracy was significantly lower in SCC than in other carcinomas. To the best of our knowledge, this has never been studied in cervical cancer for neither superficial biopsy nor conization. However, this corresponded to the results of CNB in grading other tumors. A meta-analysis of the concordance of tumor grade between CNB and excision specimen in breast cancer showed pooled agreement of 71% with a kappa of 0.54 [4].

Tumor grade is a widely known histopathologic factor in cervical carcinomas whose prognostic value has been discussed. Some authors found that high tumor grade had a prognostic impact on parametrial involvement, survival and recurrence [18, 26, 27] while others failed to show this associations in cervical cancer [28, 39, 40]. Consequently, tumor grade could represent a tool for risk stratification, but its implications for treatment choice are less clear.

One limitation of the study is the fact that the number and location of cores taken were not standardized. However, we did not find any correlation between the number of CNBs taken and their accuracy in predicting LVSI or tumor grade. Furthermore, the accuracy of CNB analysed in this study could only be compared with the accuracy of other tissue sampling techniques (e.g. cervical biopsy, conization) analysed in different studies. Obviously, this comparison has its limitations, as the assessment of accuracies could vary significantly between studies.

This study sample is characterized by a high number of women with tumors staged FIGO IIB or higher (43.1%) and its findings are therefore unique as such patients are usually submitted to primary chemo-radiotherapy. In contrast, other studies investigating the accuracy of biopsy or excision specimens have been limited by the inclusion of lower tumor stages only [23–25]. Furthermore, this is to the best of our knowledge the first study showing which characteristics of both tumor and patient may have an influence on the accuracy of histopathological examination in cervical cancer.

In our opinion, CNB does not represent a technique for the primary diagnosis of cervical cancer, as ordinary cervical biopsy may be sufficient for this purpose. We believe that the main benefit of the CNB lies in the further categorization of the cervical tumor, which could be essential in those cases, where therapy is chosen based on tumor biology.

## Conclusion

CNB has excellent accuracy in predicting histologic tumor type in cervical carcinomas. The performance in predicting LVSI and tumor grading is comparable to data published for

diagnostic conization and superficial cervical biopsy. Additionally, we show that strong peritumoral inflammation negatively affects LVSI-assessment while SCC-histology impairs tumor grading. We therefore advocate that CNB is a reasonable alternative for histopathological classification of cervical cancers if diagnostic conization is not mandatory for the staging of early tumors.

## Supporting information

**S1 Data.**
(XLSX)

## Author Contributions

**Conceptualization:** Massimiliano Lia, Lars-Christian Horn, Michael Höckel, Bahriye Aktas, Benjamin Wolf.

**Data curation:** Massimiliano Lia, Paulina Sodeikat, Benjamin Wolf.

**Formal analysis:** Lars-Christian Horn, Benjamin Wolf.

**Funding acquisition:** Michael Höckel, Benjamin Wolf.

**Investigation:** Massimiliano Lia, Paulina Sodeikat, Benjamin Wolf.

**Methodology:** Massimiliano Lia, Lars-Christian Horn, Bahriye Aktas, Benjamin Wolf.

**Software:** Massimiliano Lia, Benjamin Wolf.

**Supervision:** Lars-Christian Horn, Michael Höckel, Bahriye Aktas, Benjamin Wolf.

**Visualization:** Massimiliano Lia.

**Writing – original draft:** Massimiliano Lia.

**Writing – review & editing:** Lars-Christian Horn, Michael Höckel, Bahriye Aktas, Benjamin Wolf.

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
