## [Decision Letter · Decision Letter 0]

19 Jul 2021

PONE-D-21-05840

The diagnostic value of core needle biopsy in cervical cancer: a retrospective analysis

PLOS ONE

Dear Dr. Lia,

Thank you for submitting your manuscript to PLOS ONE. After careful consideration, we feel that it has merit but does not fully meet PLOS ONE’s publication criteria as it currently stands. Therefore, we invite you to submit a revised version of the manuscript that addresses the points raised during the review process.

We look forward to receiving your revised manuscript.

Kind regards,

Giovanni Delli Carpini

Academic Editor

PLOS ONE

Journal Requirements:

2.  Please state whether patients provided written informed consent in the ethics statement in the manuscript Methods."

3. In your Methods section, please provide additional information about the participant recruitment method and the demographic details of your participants. Please ensure you have provided sufficient details to replicate the analyses such as the recruitment date range (month and year).

5. We noted in your submission details that a portion of your manuscript may have been presented or published elsewhere.

"The abstract of this manuscript has been submitted to the 2021 ASCO Annual Meeting. The abstract submitted to the 2021 ASCO Annual Meeting is slightly different, due to different formatting guidelines, compared with the abstract of the manuscript herby submitted to PLOS ONE. However, title, authors and results are exactly the same, making it clear for every reader that the two abstracts describe the same study. Furthermore, the ASCO guidelines allow the submission of a pre-published abstract. Finally, we feel the whole study should be accessible, since the abstract alone can´t possibly reflect all aspects of the results of this study. "

Please clarify whether this conference proceeding was peer-reviewed and formally published. If this work was previously peer-reviewed and published, in the cover letter please provide the reason that this work does not constitute dual publication and should be included in the current manuscript.

6. Thank you for stating the following in the Competing Interests section:

"I have read the journal's policy and the authors of this manuscript have the following

competing interests: MH and BW received payments from the non-profit-organization

“Stiftung gynäkologische Onkologie”.

BA received honoraria from Pfizer Inc, Roche AG, Novartis AG, AstraZeneca PLC,

Amgen Inc and Daiichi Sankyo Co Ltd.

LCH, PS and ML have no conflicts to disclose."

Reviewers' comments:

Reviewer's Responses to Questions

**Comments to the Author**

1. Is the manuscript technically sound, and do the data support the conclusions?

Reviewer #1: Yes

2. Has the statistical analysis been performed appropriately and rigorously? 

Reviewer #1: Yes

3. Have the authors made all data underlying the findings in their manuscript fully available?

Reviewer #1: Yes

4. Is the manuscript presented in an intelligible fashion and written in standard English?

Reviewer #1: Yes

5. Review Comments to the Author

Reviewer #1: Overall interesting study on the use of CNB in newly diagnosed cervical cancer treated with primary surgery. The concept of having pre-operative tumor information is important and the study adds new data to the literature. However, some comments are due:

Abstract/Introduction

- LVSI and grade have a prognostic impact, but can the Authors clarify how this can impact surgery and therapeutic decision?

- Can the Authors clarify what is the primary endpoint of the study? How was the “performance” assessed?

- Authors at Line 70 state that “no studies evaluating the accuracy of CNB in cervical cancer have been published”; however, an initial description of such technique has been reported in the following study: Mascilini F, Quagliozzi L, Moro F, et al. Role of transvaginal ultrasound-guided biopsy in gynecology. Int J Gynecol Cancer. 2020;30(1):128-132. doi:10.1136/ijgc-2019-000734

Methods

- Authors should clearly state whether the study was prospective or retrospective as it is not clear in the methodology

- Can the Authors specify it in the methods the criteria to take more than one core needle biopsy for each patient?

- Line 82: was classification from FIGO 2018 or 2009 used? Can the Authors clarify in the text?

Discussion

- Line 214: Authors state that CNB can be useful to determine peri neural infiltration, but they did not report it in the results and they did not report reference for this statement

- Discussion could be shortened to 2/2.5 pages.

6. PLOS authors have the option to publish the peer review history of their article (what does this mean?). If published, this will include your full peer review and any attached files.

Reviewer #1: **Yes: **Nicolò Bizzarri

---

## [Author Response · Author response to Decision Letter 0]

12 Oct 2021

Dear editor, dear reviewer,

thank you very much for your supportive feedback which consistently improves this article. 

We have integrated the changes proposed by you in the manuscript and, as requested, an answer to every point raised by you has been added to this rebuttal letter (answers are written in bold) and some are also addressed in the revised cover letter. 

Done

2. Please state whether patients provided written informed consent in the ethics statement in the manuscript Methods."

Line 93

3. In your Methods section, please provide additional information about the participant recruitment method and the demographic details of your participants. Please ensure you have provided sufficient details to replicate the analyses such as the recruitment date range (month and year).

Recruitment: Line 94-98

Details partecipants: Line 159-166 and Table 1

The sharing of raw data, even anonymized, is not supported by the ethical committee of the university hospital in Leipzig (Stephanstraße 9A.1, 04103 Leipzig, Germany), as it contains sensitive patient information including medical history and oncological details. Thus, sharing of raw data was not part of the written consent form signed by the patients. 

Done

Minimal anonymized data has been uploaded

5. We noted in your submission details that a portion of your manuscript may have been presented or published elsewhere.

"The abstract of this manuscript has been submitted to the 2021 ASCO Annual Meeting. The abstract submitted to the 2021 ASCO Annual Meeting is slightly different, due to different formatting guidelines, compared with the abstract of the manuscript herby submitted to PLOS ONE. However, title, authors and results are exactly the same, making it clear for every reader that the two abstracts describe the same study. Furthermore, the ASCO guidelines allow the submission of a pre-published abstract. Finally, we feel the whole study should be accessible, since the abstract alone can´t possibly reflect all aspects of the results of this study. "

Please clarify whether this conference proceeding was peer-reviewed and formally published. If this work was previously peer-reviewed and published, in the cover letter please provide the reason that this work does not constitute dual publication and should be included in the current manuscript.

It has been included in the revised cover letter. 

6. Thank you for stating the following in the Competing Interests section:

"I have read the journal's policy and the authors of this manuscript have the following

competing interests: MH and BW received payments from the non-profit-organization

“Stiftung gynäkologische Onkologie”.

BA received honoraria from Pfizer Inc, Roche AG, Novartis AG, AstraZeneca PLC,

Amgen Inc and Daiichi Sankyo Co Ltd.

LCH, PS and ML have no conflicts to disclose."

It has been included in the revised cover letter. 

Reviewers' comments:

Abstract/Introduction

- LVSI and grade have a prognostic impact, but can the Authors clarify how this can impact surgery and therapeutic decision?

The role of LVSI in therapeutic decision making has been clarified in line 261-267. The role of grade is less clear, but evidence has been discussed in line 280-284. 

- Can the Authors clarify what is the primary endpoint of the study? How was the “performance” assessed?

The performance of histological subtype was performed by sensitivity and specificity

For lymphovascular space invasion and tumor-grading we calculated the sensitivity, specificity, accuracy, positive/negative predictive value and the kappa-value (chance corrected agreement between CNB and hysterectomy-specimen). This has been clarified in the methods section (line 136-144).

- Authors at Line 70 state that “no studies evaluating the accuracy of CNB in cervical cancer have been published”; however, an initial description of such technique has been reported in the following study: Mascilini F, Quagliozzi L, Moro F, et al. Role of transvaginal ultrasound-guided biopsy in gynecology. Int J Gynecol Cancer. 2020;30(1):128-132. doi:10.1136/ijgc-2019-000734

You are absolutely right and we have included this article as references in the introduction (line 69-72). 

Methods

- Authors should clearly state whether the study was prospective or retrospective as it is not clear in the methodology

This has been included in the methodology (line 77). 

- Can the Authors specify it in the methods the criteria to take more than one core needle biopsy for each patient?

The number of biopsies were determined in order to ensure a proper tissue sample for a complete histological classification of the tumor. Usually, the number of biopsies performed was higher than one. 

- Line 82: was classification from FIGO 2018 or 2009 used? Can the Authors clarify in the text?

The classification of 2009. We included this in the text (line 102).

Discussion

- Line 214: Authors state that CNB can be useful to determine peri neural infiltration, but they did not report it in the results and they did not report reference for this statement

This is a general statement about the CNB not referring exclusively to cervical cancer. Perineural infiltration was not analysed in this study, as it is not routinely assessed in CNB-tissue in cervical cancer. The reviewer is completely right to criticize the lack of a proper citation or study data backing this statement. Thus, we deleted this part as it is superfluous for this study. 

- Discussion could be shortened to 2/2.5 pages.

Done.

Done

---

## [Decision Letter · Decision Letter 1]

21 Dec 2021

The diagnostic value of core needle biopsy in cervical cancer: a retrospective analysis

PONE-D-21-05840R1

Dear Dr. Lia,

We’re pleased to inform you that your manuscript has been judged scientifically suitable for publication and will be formally accepted for publication once it meets all outstanding technical requirements.

Kind regards,

Giovanni Delli Carpini

Academic Editor

PLOS ONE

Reviewers' comments:

Reviewer's Responses to Questions

**Comments to the Author**

1. If the authors have adequately addressed your comments raised in a previous round of review and you feel that this manuscript is now acceptable for publication, you may indicate that here to bypass the “Comments to the Author” section, enter your conflict of interest statement in the “Confidential to Editor” section, and submit your "Accept" recommendation.

Reviewer #1: All comments have been addressed

2. Is the manuscript technically sound, and do the data support the conclusions?

Reviewer #1: Yes

3. Has the statistical analysis been performed appropriately and rigorously? 

Reviewer #1: Yes

4. Have the authors made all data underlying the findings in their manuscript fully available?

Reviewer #1: Yes

5. Is the manuscript presented in an intelligible fashion and written in standard English?

Reviewer #1: Yes

6. Review Comments to the Author

Reviewer #1: Overall interesting study on the use of CNB in newly diagnosed cervical cancer treated with primary surgery. The concept of having pre-operative tumor information is important and the study adds new data to the literature.

Thanks for the answers provided.

The manuscript has improved.

7. PLOS authors have the option to publish the peer review history of their article (what does this mean?). If published, this will include your full peer review and any attached files.

Reviewer #1: **Yes: **Nicolo Bizzarri

---

## [Editor Report · Acceptance letter]

26 Dec 2021

PONE-D-21-05840R1 

The diagnostic value of core needle biopsy in cervical cancer: a retrospective analysis 

Dear Dr. Lia:

I'm pleased to inform you that your manuscript has been deemed suitable for publication in PLOS ONE. Congratulations! Your manuscript is now with our production department. 

Kind regards, 

on behalf of

Dr. Giovanni Delli Carpini 

Academic Editor

PLOS ONE